# Transcriptome Analysis Reveals Differentially Expressed Genes Involved in Aluminum, Copper and Cadmium Accumulation in Tea ‘Qianmei 419’ and ‘Qianfu 4’

**DOI:** 10.3390/plants12132580

**Published:** 2023-07-07

**Authors:** Xinzhuan Yao, Hufang Chen, Baohui Zhang, Litang Lu

**Affiliations:** 1Institute of Plant Health & Medicine, College of Tea Sciences, Guizhou University, Guiyang 550025, China; xzyao@gzu.edu.cn (X.Y.);; 2Guizhou Academy of Agricultural Sciences, Guiyang 550006, China

**Keywords:** *Camellia sinensis*, transcriptome, Al, Cu and Cd content, DEGs

## Abstract

Tea, as a global nonalcoholic beverage, is widely consumed due to its economic, health and cultural importance. Polyploids have the ability to solve the problems of low yield, cold resistance and insect resistance in tea tree varieties. However, the response mechanism to aluminum and heavy metal remains unclear. In this study, the content of Al, Cu and Cd were measured in the leaves and roots of ‘Qianmei 419’ and ‘Qianfu 4’, respectively. The content of Al, Cd and Cu in the roots of the ‘Qianmei 419’ tea variety were significantly higher than in ‘Qianfu 4’ roots. Only the content of Cu in the leaves of the ‘Qianmei 419’ tea variety was significantly higher than that in the roots of the ‘Qianfu 4’ tea variety. Moreover, we found that the content of Al, Cu and Cd in the soil around the root of ‘Qianfu 4’ were higher than in the soil around the root of ‘Qianmei 419’. RNA-seq was performed to identify the DEGs involved in the accumulation of Al, Cu and Cd between ‘Qianmei 419’ and ‘Qianfu 4’. A total of 23,813 DEGs were identified in the triploid tea variety, including 16,459 upregulated DEGs and 7354 downregulated DEGs. Among them, by analyzing the expression levels of some metal transporter genes, it was found that most of the metal transporter genes were downregulated in the triploid tea plants. In short, through the analysis of transcriptome data and metal content, it was found that changes in metal transporter gene expression affect the accumulation of metals in tea plants. These results provide candidate genes to enhance multi-metal tolerance through genetic engineering technology.

## 1. Introduction

Tea, as a global nonalcoholic beverage, is widely consumed due to its economic, health and cultural importance. *Camellia sinensis*’ metal pollution of raw tea has always been a food safety concern. Many researchers have performed research on genes related to metal stress [1,2,3]. Metals such as aluminum (Al), copper (Cu), nickel (Ni), cobalt (Co), cadmium (Cd), zinc (Zn), mercury (Hg) and arsenic (As) accumulate in the soil for a long time through human activities such as industrial waste, fertilizer, smelting and sewage treatment [4]. Aluminum and heavy metals are considered important environmental contaminants that induce a wide array of physiological responses in plants [1,2,3]. Cd contamination inhibits the rate of CO_2_ fixation, decreases chlorophyll (chl) content and depresses photosynthetic activity [5,6]. Plants grown on Cd-contaminated soil are subjected to osmotic stress by minimizing leaf relative water content and reducing transpiration rate and stomatal conductance [7]. Moreover, Cd can also decrease Fe and Zn uptake, reduce Fe and Zn concentrations and cause more severe leaf chlorosis [8,9,10,11]. Jiang et al. [12] revealed that Cd interacts with mineral nutrients (Ca, Mn, Mg, K and P). Arsenic (As), a toxic metalloid that is ranked 20th in natural abundance, is widely distributed in the environment [13]. Exposure to As (V) can cause numerous stresses in plants, such as growth inhibition and several physiological disorders of plants, and can ultimately lead to death [14,15]. Arsenic toxicity induces the production and accumulation of ROS that damage biomolecules (proteins, DNA and lipids) and eventually cause cell death [16,17]. Many studies have revealed that the induction of polyploids can increase the biomass and heavy metal accumulation quantity of hyperaccumulators.

Polyploid refers to the number of complete genomes in somatic cells reaching three or more. Plant chromosome polyploidization can promote the evolution of organisms and enhance the adaptability of species. It is one of the most important examples of plant evolutionary variation in nature. Chromosome doubling first leads to a series of changes in the plant genome, which, in turn, affect the growth and developmental systems of plants [18]. A large amount of research has shown that the release of the entire genome by polyploid plants leads to changes in the structure of the genome, resulting in re-regulation of gene expression and changes in gene expression levels in the plant genome [19]. ‘Qianfu 4’ is the M1 generation obtained from the tea seed of ‘Qianmei 419’, a vegetative clone of tea tree ‘Qianmei 419’ which was irradiated with Co^60^-r rays of 6000 LUNs in 1976 [20]. It has been produced by asexual stalk cutting and breeding of single plants. ‘Qianfu 4’ and ‘Qianmei 419’ are large-leafed types and semi-arbor, semi-developed tree types. From 2002 to 2007, Chen identified ‘Qianfu 4’ and ‘Qianmei 419’ and considered ‘Qianfu 4’ to be a triploid tea tree variety [20].

The accumulation of heavy metals in tea leaves is of concern because of its impact on tea quality. These activities lead to the accumulation of metal on the surface of the soil [4,21,22,23,24,25,26,27], Heavy metals are difficult to metabolize in the human body are quite harmful, but they can be absorbed by the roots of plants through diffusion and endocytosis, especially in acidic soils [28,29,30,31,32,33,34,35]. The absorption of metals (such as Cd and Al) by plants not only threatens the growth of plants, but also poses a hazard to the food safety of plant foods. Through the food chain, between the various nutrient levels of transfer and enrichment, the absorption of metals can also cause serious potential harm to the human body. However, some of these metals, such as Zn, Cu and Ni, are essential trace elements for plants and are cofactors for a variety of enzymes, and only cause damage if they exceed a certain limit [30,36,37,38,39]. Plants have a variety of molecular and physiological mechanisms that overcome metal stress, including complex biochemical and genomic-level processes. Plants improve tolerance through various mechanisms to prevent exposure to metals present in the soil [40]. At the same time, the transportation of metal ions in plants is critical to the degree of metal enrichment in different plant tissues [41]. The differences in the ability of different tea varieties to absorb and accumulate different metals in the same habitat are mainly due to factors such as genetic regulation [3]. After the polyploidization of tea trees, the metal content and absorption and transport of metals in the body may change with the re-regulation of gene expression and gene expression levels of the tea tree genome.

Here, we discuss how changes in the ploidy of different chromosomes affect the uptake and transport of metals in plants and describe the genes involved in these processes. On the one hand, whether there is a difference in the metal content of triploid and diploid tea trees under normal growth conditions provides a theoretical basis for tea food safety; on the other hand, the molecular mechanism of the absorption and transport of diploid and triploid metals in tea trees can enrich the theoretical basis of plant metal resistance.

## 2. Results

### 2.1. Measurements of Al, Cd, and Cu Contents in ‘Qianmei 419’ and ‘QianFu No.4’

Previous studies have revealed that the induction of polyploidy can increase the biomass and heavy metal accumulation quantity of hyperaccumulators. In this study, we found differences in phenotype between ‘Qianmei 419′ and ‘QianFu No.4’. The leaves of the ‘QianFu No.4’ triploid tea variety became larger and greener than that of the ‘Qianmei 419’ diploid tea variety (Figure 1a). We measured the contents of Al, Cd and Cu in the soil of ‘Qianmei 419’ and ‘QianFu No.4’ tea plants. We found that the content of Al, Cd and Cu in the rhizosphere soil of the triploid tea plant were 7337.12, 0.16 and 29.09 mg/kg, respectively; the content of Al, Cd and Cu in the soil samples (5 cm from the main root of the tea tree) of the diploid tea plant were 7341.41, 0.16 and 29.45 mg/kg, respectively (Figure 1b). There was no significant difference in the content of these three metals in the soil samples (5 cm from the main root of the tea tree) of triploid tea plants and diploid tea plants. However, the Cd, Al, and Cu content in the soil of ‘QianFu No.4’ (50 cm away from the tea tree) were 5.62, 7943.59, 33.28 mg/kg, and the Cd, Al, and Cu content in the soil of ‘Qianmei 419′ were 53.4, 7843.27, 32.97 mg/kg, which were higher than the heavy metal content in the soil rhizosphere. It can be inferred that the tea tree roots absorbed some heavy metals. Moreover, we further determined the content of Al and heavy metal in the leaves and roots of the ‘Qianmei 419′ diploid tea variety and the ‘QianFu No.4’ triploid tea variety. The content of heavy metal Cd in the root of triploid tea plants (7.29 mg/kg) was 18 times that in the root of diploid tea plants (0.42 mg/kg) but was not detected in the leaves (Figure 1c). The average content of Cu in the leaves and roots of triploid tea plants (13.23 mg/kg in the leaves and 17.93 mg/kg in the roots) was significantly higher than in diploid tea plants (6.93 mg/kg in the leaves and 10.04 mg/kg in roots (Figure 1d)). The content of Al in the roots of triploid tea plants (6760 mg/kg in the roots) was significantly higher than in diploid tea plants (2353.33 mg/kg in the roots) (Figure 1e), and the content of Al in the leaves of triploid tea plants (1640.00 mg/kg in leaves) was not significantly higher than in diploid tea plants (1493.33 mg/kg in leaves) (Figure 1e). These results indicate that the accumulation of Cd, Al and Cu specifically occurred in the roots of triploid tea plants.

### 2.2. RNA-Seq Analysis of ‘Qianmei 419’ and ‘QianFu No.4’ Tea Varieties

To further investigate the enrichment mechanism of Al, Cu, and Cd in the ‘QianFu No.4’ triploid tea variety, we performed a comparative analysis of the transcriptome across ‘Qianmei 419’ and ‘QianFu No.4’ tea plants using the Illumina HiSeq platform. In total, 58,155,664, 62,062,126, and 48,399,274 raw reads were obtained from triploid tea tree samples (CaS4_1, CaS4_2, CaS4_3), respectively. Diploid tea tree samples (CaS419_1, CaS419_2, CaS419_3) (Appendix A) obtained 45,512,200, 43,924,994, and 60,461,068 raw reads, respectively (Appendix A). The raw reads were filtered and assembled by the de novo assembly software Trinity. The assembled sequences were redundant, so the software TGICL was used to splice to obtain the longest non-redundant single unigene set and further statistical information about the single unigene set. The data quality is displayed in Table 1.

### 2.3. Functional Annotation

BLASTX was used to perform a similarity search to annotate a unigene against various databases. All 103,448 (100%) unigenes were annotated in at least one database. A total of 35,980 (34.78%), 67,980 (65.71%), 90,547 (87.53%), 89,933 (86.94%) and 61,318 (48%) unigenes were similar to the sequences in the COG, KEGG, NR, NT and Swissprot databases, respectively (Figure 2). The E-value threshold was 1 × 10^−5^. Based on the NR annotation results, a total of 45,820 (44.29%) unigenes were annotated in the GO database by Blast2GO based on the NR annotation results (Figure 2). The cutoff value of the E-value was 1 × 10^−6^. However, due to the lack of complete genomic sequences in tea trees, only 27,030 unigenes were co-annotated into six databases (NR, NT, Swissprot, COG, GO and KEGG), accounting for 26.13% of 103,448 unigenes (Figure 2).

### 2.4. Differentially Expressed Genes (DEG) and Enrichment Analysis

Each unigene was assembled by clean reads, and then the number of reads per unigene was calculated by RSEM. The effective length of each unigene and the number of reads on the total alignment were combined, and the expression of each unigene was normalized by the FPKM method. As a result, 16,459 single genes were filtered as upregulated genes in triploid samples, and 7354 genes were calculated as downregulated genes with a cutoff of |log2(foldchange)| > 1 and padj < 0.05 by comparative analysis (Figure 3a).

To characterize the functions of the DEGs, GO enrichment analysis was performed to categorize potential functions. The DEGs were significantly annotated in the top 30 terms in the GO enrichment. The GO terms included biological process (BP), cellular component (CC) and molecular function (MF). In the BP category, the two main categories of oxidation–reduction process and protein phosphorylation accounted for the highest proportion. Of these, approximately 691 DEG were annotated as related to the oxidation–reduction process, and approximately 495 DEG were annotated as related to protein phosphorylation. For the MF category, the main category of ATP binding accounted for the highest proportion of DEGs. In the CC category, the largest category was ‘integral component of membrane’. The numbers of DEG enriched in the two categories were 1327 and 2676, respectively, which accounted for 30.61% of the enriched DEG (Figure 3b).

KEGG is a utility database for understanding advanced functions and biological systems (such as cells, organisms and ecosystems), genome sequencing and other high-throughput experimental techniques generated from molecular-level information, especially large molecular data sets. In Figure 3c, there were top 30 pathways that enriched the DEGs; among these pathways, the pathways with the highest richfactor were photosynthesis-antenna proteins and photosynthesis, while the pathways with the largest DEGs enrichment were ribosome and plant–pathogen interaction, with 596 and 533 DEGs enriched, respectively.

### 2.5. DEG of Transporters Related to Cadmium (Cd), Copper (Cu) and Aluminum (Al) in Tea

In order to answer the question of the difference in the content of metal Cd, Cu and Al in the tea plant, we screened out the DEGs associated with their absorption and transport from the differentially expressed genes in the transcriptome. We compared the expression levels of DEGs in the ATM, NRAMP, PCS, PDR, RAN and ZIP related to Cd transport in ‘QianFu No.4’ tea and ‘Qianmei 419’ tea. As shown in Figure 4a, most of these genes were downregulated in ‘QianFu No.4’ tea, and only CL1218.Contig2_All (NRAMP), CL3769.Contig6_All (PCS), and CL6459.Contig2_All (ZIP) were upregulated in ‘QianFu No.4’ tea (Appendix A).

The DEGs related to Cu transport in gene families such as CHH, CCS, COPT, HMA, PAA and ZIP were different from those of Cd. Some genes were downregulated in ‘QianFu No.4’ tea, such as CL3847.Contig1_All (HMA), CL3847.Contig2_All (HMA), CL1305.Contig6_All (HMA) and other three genes: Unigene42320_All (CHH), CL19866. Contig6_All (COPT), and Unigene12824_All (COPT). In addition, 10 genes were upregulated in ‘QianFu No.4’ tea. (Figure 4b). Genes related to Al transport were mainly concentrated in the ALMT and MATE gene families and the STOP transcription factor. Among the 24 DEGs related to Al transport, only CL1274.Contig5_All (MATE) and CL6638.Contig4_All (MATE) were upregulated in ‘QianFu No.4’ tea, and the remaining DEGs were all downregulated in ‘QianFu No.4’ tea. This gene expression trend was consistent with Cd (Figure 4c).

### 2.6. Real-Time Quantitative PCR Validation

To validate the RNA-seq results in triploid and diploid tea trees, another strategy for single gene regulation was chosen. Four up- and four downregulated unigenes were selected for verification by real-time quantitative PCR (qRT-PCR) with identical RNA samples to those used for RNA-Seq (Figure 5). Although the expression level of the test gene was not completely consistent, the expression trend of the test gene was similar (R = 0.8569).

## 3. Discussion

*Camellia sinensis* is an important global natural resource and a non-alcoholic plant source of healthy drinks [42,43]. The metal content of tea trees has become one of the most concerning food safety issues [44]. Although researchers have also applied transcriptomics to discover genes related to metal stress in tea plants [3,45,46], the enrichment and transport of metals by polyploid tea plants have not been reported on yet.

In this regard, we determined the content of metals such as Cd, Al and Cu in ‘Qianfu 4’ and ‘Qianmei 419’. In order to eliminate the influence of the difference in Cd, Al and Cu content in the soil, this research tested the content of Cd, Al, Cu and other metals in the rhizosphere soil of ‘Qianfu 4’ and ‘Qianmei 419’. There was no significant difference in the content of Cd, Al and Cu in rhizosphere soil, indicating that the two tea plants grew in the same soil conditions. The content of Cd, Cu and Al in the roots of ‘Qianfu 4’ were significantly higher than in ‘Qianmei 419’. This shows that the roots of ‘Qianfu 4’ had a stronger ability to accumulate enrichment metals. These results are consistent with those of Wang Qiang [47], who discovered that polyploid plants can accumulate cadmium and store it in their roots. However, only the Cu content in the leaves of ‘Qianfu 4’ was significantly higher than that of ‘Qianmei 419’. The concentrations of Cd and Al in the leaves of ‘Qianfu 4’ were smaller, showing an opposite trend in the roots. The low metal content of tea leaves reduces tea consumption in metal-polluted environments [44].

In this study, RNA-Seq technology was used to construct the transcriptome of ‘Qianfu 4’ and ‘Qianmei 419’ leaves. We aimed to obtain complete and accurate comparative transcriptome information of ‘Qianfu 4’ and ‘Qianmei 419’ to reveal the regulatory mechanism and related genes that reduce the metal enrichment degree of ‘Qianfu 4’ tea leaves. RNA-seq is one of the most famous transcriptome research methods. It is sufficient for researchers to understand expression patterns and identify genes [48]. In this study, a total of 23,813 transcripts were identified as differentially expressed genes (DEGs). The qRT-PCR expression level was basically consistent with the change in the transcript abundance identified by RNA-seq, indicating that the RNA-seq data in this study are credible. Through statistical analysis of the terms and pathways of enriched GO related to DEG, we found that the transcripts in triploid tea plants had the most significant and frequently enriched terms and pathways, which are mainly involved in the integral component of membrane of the cellular component. There were also many DEGs involved in the biological oxidation–reduction process. Plants develop a variety of metal defense, transport, absorption and other biochemical strategies and mechanisms, all related to cell membranes [49], and the presence of excessive heavy metals and aluminum can increase membrane lipid peroxidation and reduce cell membrane stability [50].

In addition, the KEGG enrichment results of differentially expressed genes (DEGs) showed that the two pathways of photosynthesis-antenna proteins and photosynthesis had the highest degree of enrichment. Plants are highly sensitive to metal ions [51]. Among them, Cd and Cu have multidirectional effects on photosynthesis [52]. Unlike Cd, Cu is an essential plant trace element. Cu participates in photosynthetic electron transport, is a component of plastocyanin and is also important in regulating the function of PSII [53]. This shows that photosynthesis is closely related to the content of metal ions in plants, which is one of the reasons why it is possible to enrich photosynthesis.

The transport of metal ions in plants is critical to the degree of metal enrichment in different plant tissues [41], and metal transporters are responsible for metal transport between plant tissues. For example, the heavy-metal ATPase (HMA) family plays an important role in transition metal transport in plants [54]. Moreover, in 2010, Mills found that Arabidopsis AtHMA4 can transport Cd^2+^ [55]. Some transcription factors are also involved in the transcription of metal ions. For example, STOP plays a critical role in tolerance of major stress factors in acid soils, such as proton H+ and aluminum ion Al^3+^, which are required for the expression of genes in response to acidic stress (e.g., ALMT1 and MATE) and Al-activated citrate exudation [56,57,58]. Therefore, we counted the DEGs expressed in some metal transporters and transcription factors (Figure 4). Among them, 25 DEGs related to Cd transport were identified (CL1775.Contig2_All, CL4038.Contig2_All, CL3769.Contig6_All, CL4653.Contig1_All, CL3134.Contig5_All, CL6452.Contig7_All, etc.), of which 22 DEGs were downregulated in ‘Qianfu 4’ and 3 DEGs were upregulated in ‘Qianfu 4’. In addition, 16 DEGs related to Cu transport were identified (Unigene42320_All, CL18886.Contig2_All, CL19866.Contig2_All, etc.), of which 6 DEGs were downregulated in ‘Qianfu 4’ and 10 DEGs were upregulated in ‘Qianfu 4’. In addition, 24 DEGs related to Al transport were identified (CL10687.Contig2_All, CL11478.Contig3_All, CL10157.Contig1_All, etc.), of which 22 DEGs were downregulated in ‘Qianfu 4’ and 2 DEGs were upregulated in ‘Qianfu 4’. This indicates that most of the genes related to Cd and Al transport are downregulated in ‘Qianfu 4’, reducing the content of Cd and Al transported to leaves in triploid roots, while the number of genes related to Cu transport is upregulated in ‘Qianfu 4’. Moreover, this may be the reason why the Cu content in triploid leaves is higher than that in the ‘Qianmei 419’ leaves.

In summary, ‘Qianfu 4’ can accumulate more Cd, Cu and Al. However, the downregulation of transport genes related to Cd and Al caused no significant difference in the content of Cd and Al in triploid leaves. This phenomenon may be beneficial to planting polyploid tea trees in areas contaminated with heavy metals and Al and keeping the quality of tea leaves unchanged. Moreover, this molecular mechanism of transport can enrich the theoretical basis of plant metal resistance.

## 4. Materials and Methods

### 4.1. Plant Materials and Growth Conditions

The tea tree ‘QianMei 419’ is a small- and medium-sized-leaf breed cultivated by the Tea Science Institute of Guizhou Academy of Agricultural Sciences (Guiyang, China). ‘Qianfu 4’ was obtained using Co60-γ rays to mutate the seeds of ‘QianMei 419’ and was strictly self-pollinated by this institute. Chen et al. [20] studied mitosis in the leaves of ‘QianFu No.4’ and found 45 chromosomes, which is 1 more group than the number in ‘QianMei 419’. For this study, ‘QianMei 419’ and ‘QianFu No.4’ were planted in a tea plantation through cuttings in 2015 (N 27.76, E 107.49, altitude 760 m a.s.L.) (Figure 1). We selected ‘QianMei 419’ and ‘QianFu No.4’ material that grew rapidly and uniformly and that was not infected with pathogens. The leaves and roots of the ‘Qianmei 419’ and ‘QianFu No.4’ tea varieties were collected for the determination of aluminum and heavy metal content. All samples were collected and immediately put into liquid nitrogen. At the same time, soil samples (5 cm from the main root of the tea tree and 50 cm away from the tea tree) were taken: S-shaped distribution points are used in flat tea gardens, and the central position of the tea tree and fertilization ditch was taken. The surface dead branches and fallen leaves were removed and the soil layer was sampled in the 10 cm soil layer using a military shovel. Each mixed-soil sample needs to be collected from 15–20 sample points. After thorough mixing, 500 g of soil samples were retained using the four-part method for detecting heavy metal content.

### 4.2. RNA Extraction and Sequencing

Four tissues (root and leaf from diploid ‘Qianmei 419’ and triploid ‘QianFu No.4’) were used to perform RNA-seq analysis. Each sample processed three biological replicates. Total RNA was isolated from samples using an RNA extraction kit (Tiangen, Beijing, China) according to the manufacturer’s protocol. Illumina RNA-Seq libraries were prepared and sequenced on an Illumina Novaseq 6000 (san Diego, CA, USA) following the manufacturer’s instructions. The sequencing quality of RNA-Seq data was assessed by FastQC software (1.11.4). After removing low-quality reads through SOAPnuke software (2.1), clean reads from each sample were aligned to the Chayote genome by STAR software with default parameters. FPKM (Fragments Per Kilobase per Million) was used to calculate the gene expression values. Htseq software was used to analyze gene expression. The DESeq program was used to perform DEG analysis among the different samples.

### 4.3. Functional Annotation and Enrichment Analyses

The assembled unigenes that might putatively encode proteins were used to blast against the seven databases—nr (http://www.ncbi.nlm.nih.gov/) (accessed on 23 June 2022), nt (NCBI nucleotide sequences), Pfam (Protein family, http://pfam.sanger.ac.uk/) (accessed on 23 June 2022), KOG (euKaryotic Ortholog Groups, http://www.ncbi.nlm.nih.gov/cog/) (accessed on 23 June 2022), Swiss-Prot (a manually annotated and reviewed protein sequence database, http://www.expasy.ch/sprot/) (accessed on 23 June 2022), KEGG (Kyoto Encyclopedia of Genes and Genomes, http://www.genome.jp/kegg/) (accessed on 24 June 2022)and GO (Gene Ontology, http://www.geneontology.org/) (accessed on 24 June 2022)—using the BLASTX algorithm. In addition, clusterProflier was used to perform GO and KEGG enrichment analysis to cluster differentially expressed genes (DEG) into different modules.

### 4.4. Determination of Al, Cu and Cd Content in Root and Leaf

About 0.2~1 g of test sample was placed into the Teflon digestion tank (accurate to 0.1 mg) while adding 5 mL of nitric acid. When the reactions finished, the tanks were sealed with caps and placed into a microwave digestion instrument (WX-8000) (Shanghai Hao Instrument Technology Development Co., Ltd., Shanghai, China). The digestion tanks were taken out and placed in a fume hood with a temperature below 50 °C, which was rinsed with ultrapure water 3~4 times. Then, the tanks were transferred it into a 50 mL volumetric flask, rinsed, diluted to volume with ultrapure water, and the contents of Cu, Al and Cd were measured by inductively coupled plasma mass spectrometry (ICP-MS, Thermo Fisher Scientific, Waltham, MA, USA).

### 4.5. qRT-PCR Analysis

To verify the accuracy of RNA-Seq results, 10 DEGs were randomly selected for qRT-PCR analysis. Total RNA was extracted from leaf samples of ‘QianFu No.4’ and ‘Qianmei 419’ with RNAiso Plus (TaKaRa, Tokyo, Japan). Then, the samples were treated with RNAase-free DNase I (RQ1RNase-Free DNase; Promega, Madison, WI, USA) to remove genomic DNA, after which first-strand cDNA was synthesized (Revert Aid First-strand cDNA Synthesis Kit; Fermentas, Burlington, ON, Canada). The qRT-PCR reactions utilized for gene expression analysis were conducted in a 7500 Fast Real-Time PCR instrument (Applied Biosystems, Waltham, MA, USA) using SYBR Premix Ex Taq (TaKaRa Biotech Co., Ltd., Dalian, China) according to the manufacturer’s instructions. The amplification cycling conditions were as follows: 5 min at 95 °C, 15 s at 95 °C, and 1 min at 60 °C for 39 cycles. Three biological replicates were performed. EF1 gene was used as an internal control for normalization, and relative gene expression was quantified using the 2^−ΔΔCt^ method. Gene-specific qRT-PCR primers were designed using primer premier 5.0 software (Appendix A).

## 5. Conclusions

In this study, a large number of DEGs were obtained from the RNA-seq data of tea plants, which provide more opportunities for studying the molecular regulatory mechanisms of cadmium, aluminum and copper stress responses. In transcriptome data, we identified 25 DEGs related to cadmium transport, 16 DEGs related to Cu transport, and 24 DEGs related to Al transport. This indicates that most of the genes related to Cd and Al transport were downregulated in ‘QianFu No.4’, resulting in a decrease in the content of Cd and Al transported to the leaves in the triploid root, while the genetic genes related to Cu transport were upregulated in ‘QianFu No.4’. In addition, this may be the reason why the copper content in the triploid leaves was higher than that in ‘Qianmei 419’ leaves. These results will help us understand the mechanisms of cadmium, aluminum and copper stress responses in tea plants and screen key candidate genes for future molecular breeding.

## Figures and Tables

**Figure 1 plants-12-02580-f001:**
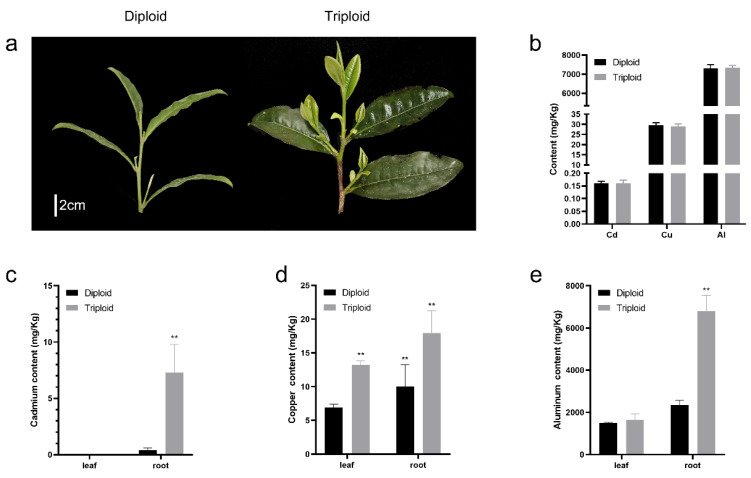
(**a**) The morphological structure of the leaves of ‘Qianmei 419’ (diploid) and ‘QianFu No.4’ (triploid); (**b**) The contents of Cd, Cu and Al in the rhizosphere soil of ‘Qianmei 419’ (diploid) and ‘QianFu No.4’ (triploid); (**c**) The content of Cd in the leaves and roots of ‘Qianmei 419’ (diploid) and ‘QianFu No.4’ (triploid); (**d**) The content of Cu in the leaves and roots of ‘Qianmei 419’ (diploid) and ‘QianFu No.4’ (triploid); (**e**) The content of Cd in the leaves and roots of ‘Qianmei 419’ (diploid) and ‘QianFu No.4’ (triploid). Data are means ± SE (n = 3). Asterisks indicate significant differences (** *p* < 0.01 by Duncan’s test).

**Figure 2 plants-12-02580-f002:**
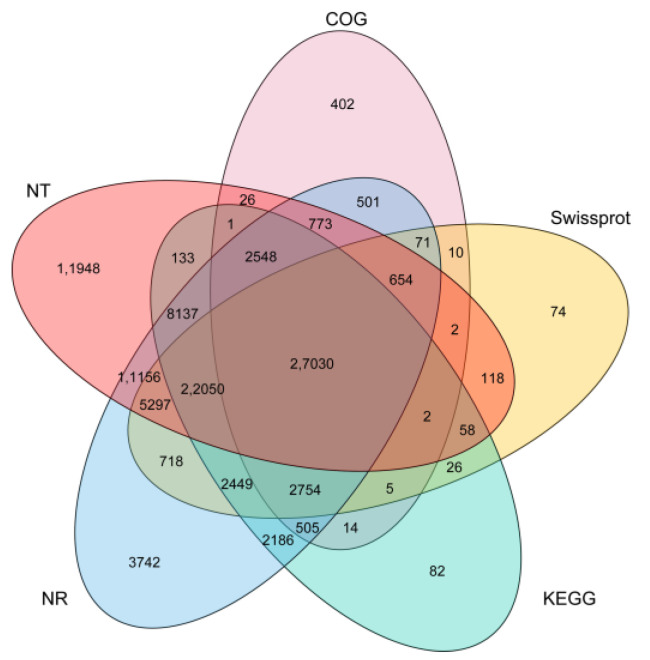
Venn diagram of NR, NT, COG, KEGG and Swissprot function annotations.

**Figure 3 plants-12-02580-f003:**
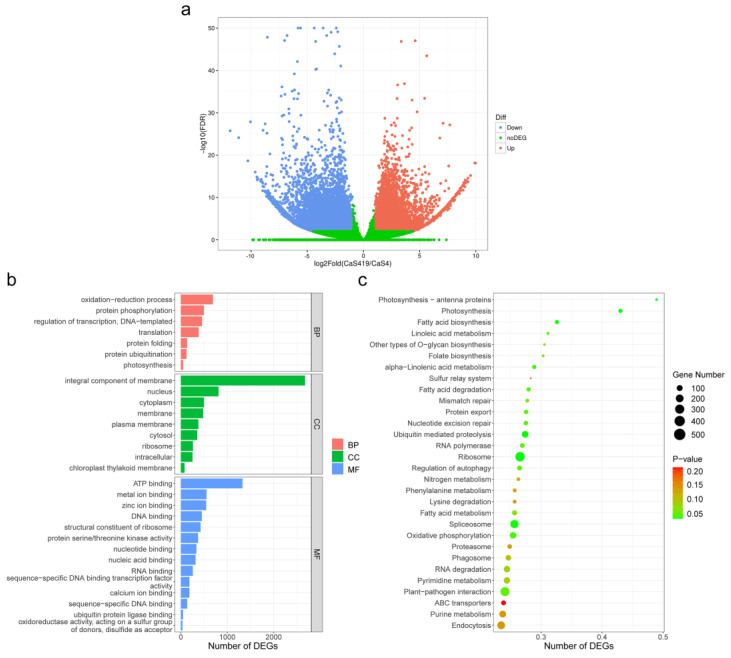
(**a**) Expression patterns of differentially expressed genes (DEGs) identified between ‘QianFu 4’ triploid tea and ‘Qianmei 419’ diploid tea. Red(16,459 single genes) and blue(7354 genes) dots represent DEGs, while green dots indicate genes that were not differentially expressed. (**b**) GO enrichment results of DEGs. Red represents a biological process, blue represents the cellular component and cyan represents molecular function. (**c**) KEGG enrichment results of DEGs.

**Figure 4 plants-12-02580-f004:**
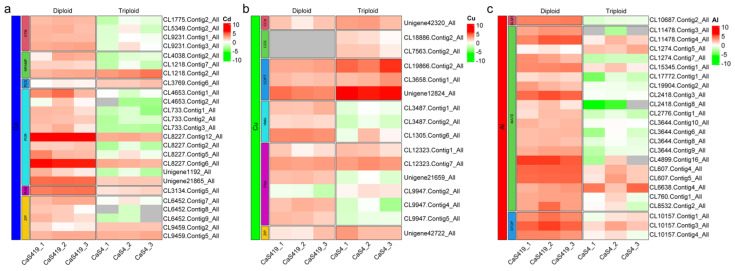
Expression profile of DEGs related to Cd, Cu and Al. (**a**) Expression profile of DEGs related to Cd, (**b**) expression profile of DEGs related to Cu, (**c**) expression profile of DEGs related to Al. Expression profiles are Log2 (FPKM).

**Figure 5 plants-12-02580-f005:**
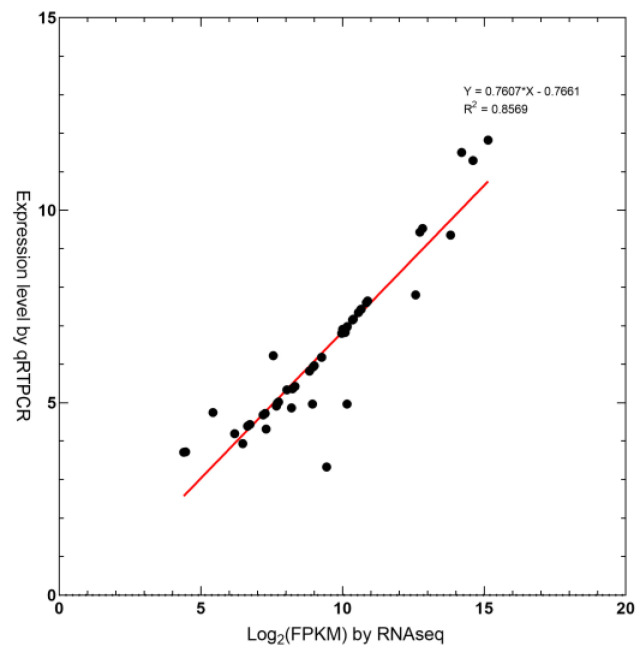
Validation of RNA-seq results by qRT−PCR for eight genes.

**Table 1 plants-12-02580-t001:** De novo assembly result statistics.

Assembly Feature	Value
Total_Number	154,097
Min_Length	200
Max_length	20,083
Mean_Length	825.67
N50	1349
N90	332
GC	0.411

## Data Availability

The data presented in this study are available on request from the corresponding author.

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
