# Peer review of "Transcriptome Analysis Reveals Differentially Expressed Genes Involved in Aluminum, Copper and Cadmium Accumulation in Tea ‘Qianmei 419’ and ‘Qianfu 4’"

_plants, 2023, doi:10.3390/plants12132580_

Round 1

Reviewer 1 Report

The manuscript entitled “Transcriptome Analysis Reveals Differentially Expressed involved in aluminum, copper and cadmium Accumulation in Tea‘Qianmei 419’ and ‘Qianfu 4’” to me seems to be an interesting study wherein authors have compared the accumulation of Al, Cu, and Cd in two tea varieties: ‘Qianmei 419’ (triploid) and ‘Qianfu 4’ (diploid) and conducted the RNA-seq to study the effect of metal tolerance in tea plants. The methodology adopted in the study technically seems to be fine, However the language needs some polishing.

·       Figure 3 & 4 will be difficult for red-green colorblindness.  Recommend using yellow-blue scale, which is now recommended.   In reference to Figure 5, to provide a bar graph instead of heatmap along with their significance level, Also authors should also include a correlation between qRT and RNASeq data to check how much concordance is it.

·       Page#3, Line#103 to 105 needs to be rephrased.

·       There are some spacing typos in the manuscript as instance Line#122 space need in between “indicate significantly”; Line No. 293 “strictly self-pollinated” which should be corrected across the current version of the manuscript.

OK 

Author Response

ar  Reviewer

Thank you for providing us with this great opportunity to submit a revised version of our manuscript. We appreciate the detailed and constructive comments provided by the reviewers. We have carefully revised the manuscript by incorporating all the suggestions by the review panel.

We hope that this revised manuscript has addressed your concerns and look forward to hearing from you.

Sincerely,

The Authors:Xinzhuan Yao ,Hufang Chen Baohui Zhang and Litang Lu

Reviewer: 1

Thank you very much for your time involved in reviewing the manuscript and your very encouraging comments on its merits.

Comments:

  1. Figure 3 & 4 will be difficult for red-green colorblindness.  Recommend using yellow-blue scale, which is now recommended.   In reference to Figure 5, to provide a bar graph instead of heatmap along with their significance level, Also authors should also include a correlation between qRT and RNASeq data to check how much concordance is it.

Response 1:

Thank you for your extremely helpful suggestion on improving the accessibility of our manuscript. To address your first comment that there is Figure 3 is an image provided by the sequencing process company that is difficult to modify, but it is acceptable for most authors. Thanks for your understanding. Figures 4 and 5 have been replaced with the latest version according to expert opinions in the manuscript.

  1. Page#3, Line#103 to 105 needs to be rephrased.

Response 2:

Thank you for your extremely helpful suggestion on improving the accessibility of our manuscript. To address your first comment that we found the differences in phenotype between ‘Qianmei 419’ and ‘QianFu No.4.’ The leaves of the ‘QianFu No.4’ triploid tea variety became larger and greener than that of the ‘Qianmei 419’ diploid tea variety (Figure1a). We measured the content of Al, Cd and Cu in the rhizosphere soil of ‘Qianmei 419’ and ‘QianFu No.4’ tea plants. We found that the contents of Al, Cd and Cu in the rhizosphere soil of the triploid tea plant were 7337.12, 0.16 and 29.09 mg/kg, respectively; the contents of Al, Cd and Cu in the rhizosphere soil of the diploid tea plant were 7341.41, 0.16 and 29.45 mg/kg, respectively (Figure1b). There was no significant difference in the content of these three metals in the rhizosphere soil of triploid tea plants and diploid tea plants. Moreover, we further determined the content of Al and heavy metal in the leaves and roots of the ‘Qianmei 419’ diploid tea variety and the ‘QianFu No.4’ triploid tea variety. The content of heavy metal Cd in the root of triploid tea plants (7.29mg/kg) was 18 times that in the root of diploid tea plants (0.42mg/kg) but was not detected in leaves (Figure1c). The average content of Cu in the leaves and roots of triploid tea plants (13.23 mg/kg in the leaves and 17.93 mg/kg in the roots) was significantly higher than that of diploid tea plants (6.93 mg/kg in the leaves and 10.04 mg/kg in roots (Figure 1d). The content of Al in the roots of triploid tea plants (6760 mg/kg in the roots) was significantly higher than that of diploid tea plants (2353.33 mg/kg in the roots) (Figure 1e), and the content of Al in the leaves of triploid tea plants (1640.00 mg/kg in leaves) was not significantly higher than that of diploid tea plants (1493.33 mg/kg in leaves) (Figure 1e). These results indicate that the accumulation of Cd, Al and Cu specifically occurred in the roots of triploid tea plants.

  1. There are some spacing typos in the manuscript as instance Line#122 space need in between “indicate significantly”; Line No. 293 “strictly self-pollinated” which should be corrected across the current version of the manuscript.

 Response 3:

Thank you for your extremely helpful suggestion on improving the accessibility of our manuscript. To address your first comment that Asterisks indicate significant differences (*p<0.05 and **p<0.01 by Duncan’s test) modified to Asterisks indicate significant differences (*p <0.05 and **p <0.01 by Duncan’s test) ï¼›strictly selfpollinated modified to strictly self-pollinated.

Reviewer 2 Report

In this study, some quantities of DEG obtained from RNA-seq data of tea plants, for the study of the molecular regulatory mechanisms of cadmium, aluminum, and copper stress responses are shown. In transcriptome data 25 DEGs related to cadmium transport, 16 DEGs related to Cu transport, and 24 DEGs related to Al transport were explained and identified. most of the genes related to Cd and Al transport are downregulated in‘QianFu No. 4’, resulting in a decrease in the content of Cd and Al transported to the leaves in the triploid root, while the genetic genes related to Cu transport are upregulated in ‘QianFu No. 4’. In addition, this may be the reason why the copper content in the triploid leaves is higher than that in‘Qianmei 419 leaves. mechanism of cadmium, aluminum, and copper stress responses in tea plants is explanined in this paper and screen key candidate genes for future molecular breeding. Good figures including the expression profile of DEGs related to Cd,Cu and Al. a. Expression profile of DEGs related to Cd; Expression profile of DEGs related to Cu; and  Expression profile of DEGs related to Al are depicted nicely
Comments please check accurately English language.

Comments please check accurately English language.

Author Response

Dear  Reviewer

Thank you for providing us with this great opportunity to submit a revised version of our manuscript. We appreciate the detailed and constructive comments provided by the reviewers. We have carefully revised the manuscript by incorporating all the suggestions by the review panel.

We hope that this revised manuscript has addressed your concerns and look forward to hearing from you.

Sincerely,

The Authors:Xinzhuan Yao ,Hufang Chen Baohui Zhang and Litang Lu

Reviewer:

In this study, some quantities of DEG obtained from RNA-seq data of tea plants, for the study of the molecular regulatory mechanisms of cadmium, aluminum, and copper stress responses are shown. In transcriptome data 25 DEGs related to cadmium transport, 16 DEGs related to Cu transport, and 24 DEGs related to Al transport were explained and identified. most of the genes related to Cd and Al transport are downregulated in‘QianFu No. 4’, resulting in a decrease in the content of Cd and Al transported to the leaves in the triploid root, while the genetic genes related to Cu transport are upregulated in ‘QianFu No. 4’. In addition, this may be the reason why the copper content in the triploid leaves is higher than that in‘Qianmei 419’ leaves. mechanism of cadmium, aluminum, and copper stress responses in tea plants is explanined in this paper and screen key candidate genes for future molecular breeding. Good figures including the expression profile of DEGs related to Cd,Cu and Al. a. Expression profile of DEGs related to Cd; Expression profile of DEGs related to Cu; and  Expression profile of DEGs related to Al are depicted nicely

 Response

Thank you for your extremely helpful suggestion on improving the accessibility of our manuscript. To address your first comment that the Quality of English Language

Reviewer 3 Report

The paper must be greatly improved. The authors should ask a professional soil scientist during revision. Nonsense sentences as L 69- Metals are difficult to degrade in soils- may be then avoided. 

The abstract and intro must be fully revised. The first sentence in the abstract and intro is full of nonsense.

Al, as well as Fe is abundant in soils, as part of silicates or Al/Fe-oxides and strongly differs from metals such as Cd, Hg, As.

No description of the recovery of rhizosphere soil is given.   

The English must be improved.

Author Response

Dear  Reviewer

Thank you for providing us with this great opportunity to submit a revised version of our manuscript. We appreciate the detailed and constructive comments provided by the reviewers. We have carefully revised the manuscript by incorporating all the suggestions by the review panel.

We hope that this revised manuscript has addressed your concerns and look forward to hearing from you.

Sincerely,

The Authors:Xinzhuan Yao ,Hufang Chen Baohui Zhang and Litang Lu

Reviewer

The paper must be greatly improved. The authors should ask a professional soil scientist during revision. Nonsense sentences as L 69- Metals are difficult to degrade in soils- may be then avoided. 

Response 1

 Thank you for your very helpful suggestions on improving the accessibility of our manuscript. In response to your first comment that we have made revisions to the abstract and preface in the manuscript.

The abstract and intro must be fully revised. The first sentence in the abstract and intro is full of nonsense. Al, as well as Fe is abundant in soils, as part of silicates or Al/Fe-oxides and strongly differs from metals such as Cd, Hg, As.

Response2  Thank you for your very helpful suggestions on improving the accessibility of our manuscript. In response to your first comment that we have made extensive revisions to the abstract and preface during the revision. Al soil is rich in content and is a part of silicate or Al oxide, but it is absorbed by the roots of tea trees and transported to the leaves of tea trees, resulting in excessive content and seriously affecting the quality of tea.

No description of the recovery of rhizosphere soil is given. 

Response3  Thank you for your very helpful suggestions on improving the accessibility of our manuscript. In response to your first comment that this paper mainly studies the differences in heavy metal content in tea leaves of Qianmei 419 and Qianfu 4, and the effects of related genes on metal content. The description of rhizosphere soil restoration may be studied later.

Reviewer 4 Report

Main criticism. Unless I am mistaken, as it stands, readers will not be able to identify the genes which are mentioned. The authors should make sure that a supplementary file (e.g. S2) contains an identifier leading to a publicly available gene sequence, and the corresponding gene in Arabidopsis. The sequence of the most important DEGs should be given in a supplementary file. I also suggest resubmitting larger version of figures 3 and 4. There are also a number of English language problems. Examples: Line 13: maybe ‘Polyploid cultivars addressed the ability…’ would be clearer. Line 40: ‘plants’, not ‘plantss’; ‘fixation’ Line 49: ‘finally’ Althrough the ms. the authors should check there is a space before [ref.] Line 64: ‘Chen et al.’ Line 68: what does ‘these activities” refer to? Lines 94-95: it seems part of the sentence is missing. Line 231: replace comma after ‘root’ by a . Line 237: ‘better enough’ is not appropriate. Line 282: ‘Although’ does not make sense in this sentence. Main criticism. Unless I am mistaken, as it stands, readers will not be able to identify the genes which are mentioned. The authors should make sure that a supplementary file (e.g. S2) contains an identifier leading to a publicly available gene sequence, and the corresponding gene in Arabidopsis. The sequence of the most important DEGs should be given in a supplementary file. I also suggest resubmitting larger version of figures 3 and 4. There are also a number of English language problems. Examples: Line 13: maybe ‘Polyploid cultivars addressed the ability…’ would be clearer. Line 40: ‘plants’, not ‘plantss’; ‘fixation’ Line 49: ‘finally’ Althrough the ms. the authors should check there is a space before [ref.] Line 64: ‘Chen et al.’ Line 68: what does ‘these activities” refer to? Lines 94-95: it seems part of the sentence is missing. Line 231: replace comma after ‘root’ by a . Line 237: ‘better enough’ is not appropriate. Line 282: ‘Although’ does not make sense in this sentence.

Author Response

Dear  Reviewer

Thank you for providing us with this great opportunity to submit a revised version of our manuscript. We appreciate the detailed and constructive comments provided by the reviewers. We have carefully revised the manuscript by incorporating all the suggestions by the review panel.

We hope that this revised manuscript has addressed your concerns and look forward to hearing from you.

Sincerely,

The Authors:Xinzhuan Yao ,Hufang Chen Baohui Zhang and Litang Lu

Reviewer

  1. The authors should make sure that a supplementary file (e.g. S2) contains an identifier leading to a publicly available gene sequence, and the corresponding gene in Arabidopsis. The sequence of the most important DEGs should be given in a supplementary file.

Response1 

Thank you for your extremely helpful suggestion on improving the accessibility of our manuscript. To address your first comment that there supplementary file 2

  1. I also suggest resubmitting larger version of figures 3 and 4.

Response 2

Thank you for your extremely helpful suggestion on improving the accessibility of our manuscript. To address your first comment that there is Figure 3 is an image provided by the sequencing process company that is difficult to modify, but it is acceptable for most authors. Thanks for your understanding. Figures 4 have been replaced with the latest version according to expert opinions in the manuscript.

  1. There are also a number of English language problems. Examples: Line 13: maybe ‘Polyploid cultivars addressed the ability…’ would be clearer. Line 40: ‘plants’, not ‘plantss’; ‘fixation’ Line 49: ‘finally’ Althrough the ms. the authors should check there is a space before [ref.] Line 64: ‘Chen et al.’ Line 68: what does ‘these activities” refer to? Lines 94-95: it seems part of the sentence is missing. Line 231: replace comma after ‘root’ by a . Line 237: ‘better enough’ is not appropriate. Line 282: ‘Although’ does not make sense in this sentence.

Response3  

Thank you for your extremely helpful suggestion on improving the accessibility of our manuscript. To address your first comment that there ‘The polyploids processed the ability to overcome the problem of low yield, limited rainfall and increase cold- and pest-resistance in tea growing areas’ Revised to ‘Polyploids have the ability to solve the problems of low yield, cold resistance and insect resistance in tea tree varietie’. 

ther comments will be revised in the revised draft.

Round 2

Reviewer 3 Report

The authors describe results of rhizosphere metal concentrations (fig 1b). I noted correctog the fisrt version that the recovery of rhizosphere soil will be described elsewhere. Omit the results on rhizosphere soil or describe the methods on gaining this soil in detail!

see above

Author Response

Thank you for your extremely helpful suggestion on improving the accessibility of our manuscript. To address your first comment that the experimental materials for this paper are from a tea plantation of Meitan County Academy of Agricultural Sciences in Guizhou Province, with a history of more than 5 years. ‘Qianmei 419’ and ‘Qianfu 4’ grew in the same tea plantation, with identical growth environments. The insignificant difference in heavy metal content detected in the soil can directly indicate that the growth environments of these two materials are the same. The paper mainly studies the impact and accumulation of heavy metals in soil on the quality of tea in two varieties, ‘Qianmei 419’ and ‘Qianfu 4.’. There is no research on the restoration of soil roots and whether there is soil improvement. Soil samples are taken: S-shaped distribution points are used in flat tea gardens, and the central position of the tea tree and fertilization ditch is taken. The surface dead branches and fallen leaves are removed and the soil layer is sampled in the 0-30 cm soil layer using a military shovel. Each mixed soil sample needs to be collected from 15-20 sample points. After thorough mixing, 500g of soil samples are retained using the four part method for detecting heavy metal content.

Round 3

Reviewer 3 Report

I recommended several times to describe the recovery of rhizosphere soil. The authors didn´t show any knowledge of rhizophere soil and its recovery. Rhizosphere soil is soil in close proximity to the roots (up to about 2-3 mm away from the root) and has nothing to do with the description in the submitted ms. I recommend rejection.  

see above

Author Response

This paper mainly analyzes the genes regulating the difference of heavy metals between‘QianFu No.4’ and‘Qianmei 419’ by comparing the metal content in the Transcriptome and tea, laying a foundation for subsequent research.Based on the expert's suggestions, we have added:However, the Cd, Al, and Cu content in the soil of ‘QianFu No.4’ (50 centimeters away from the tea tree) was 5.62, 7943.59, 33.28 mg/kg, and the Cd, Al, and Cu content in the soil of ‘Qianmei 419’ was 53.47843.27,32.97 mg/kg, which were higher than the heavy metal content in the soil rhizosphere. It can be inferred that the tea tree roots absorbed some heavy metals.